# MiR-1208 Increases the Sensitivity to Cisplatin by Targeting TBCK in Renal Cancer Cells

**DOI:** 10.3390/ijms20143540

**Published:** 2019-07-19

**Authors:** Eun-Ae Kim, Ji-Hoon Jang, Eon-Gi Sung, In-Hwan Song, Joo-Young Kim, Tae-Jin Lee

**Affiliations:** Department of Anatomy, College of Medicine, Yeungnam University, 170 Hyeonchung-Ro, Nam-Gu, Daegu 42415, Korea

**Keywords:** miR-1208, cisplatin, TRAIL, TBCK, Caki-1

## Abstract

MicroRNAs (miRNAs) can be used to target a variety of human malignancies by targeting their oncogenes or tumor suppressor genes. Recent evidence has shown that miRNA-1208 (miR-1208) was rarely expressed in a variety of cancer cells, suggesting the possibility that miR-1208 functions as a tumor suppressor gene. Herein, ectopic expression of miR-1208 induced the accumulation of sub-G1 populations and the cleavage of procaspase-3 and PARP, which could be prevented by pre-treatment with the pan-caspase inhibitor, Z-VAD. In addition, miR-1208 increased the susceptibility to cisplatin and TRAIL in Caki-1 cells. Luciferase reporter assay results showed that miR-1208 negatively regulates TBC1 domain containing kinase (TBCK) expression by binding to the miR-1208 binding sites in the 3′-untranslated region of TBCK. In addition, miR-1208 specifically repressed TBCK expression at the transcriptional level. In contrast, inhibition of endogenous miR-1208 by anti-miRs resulted in an increase in TBCK expression. Downregulation of TBCK induced by TBCK-specific siRNAs increased susceptibility to cisplatin and TRAIL. These findings suggest that miR-1208 acts as a tumor suppressor and targets TBCK directly, thus possessing great potential for use in renal cancer therapy.

## 1. Introduction

MicroRNAs (MiRNAs), small molecules of single-stranded RNA with a length of 21–25 nucleotides, regulate gene expression and control many physiological processes such as development, proliferation, differentiation, and apoptosis [1,2]. MiRNAs are classified as either oncogenic miRNAs that promote cancer development or tumor-suppressive miRNAs that negatively regulate oncogenes and suppress tumor development and invasion [3]. A single miRNA can alter the expression of more than 100 different genes, thus controlling an entire cancer-specific signaling pathway [4]. Abnormalities in miRNA expression have been identified in all human tumors [5]. Therefore, there has been widespread attention paid to the potential role of miRNAs in tumorigenic or tumor-suppressive processes.

Renal cell carcinoma (RCC) is a common cancer that accounts for about 3% of all human cancers [6]. The RCC morbidity rate is rapidly increasing due to the development of resistance of metastatic RCC to chemotherapy and radiotherapy over the last 20 years [7,8]. Thus, a better understanding of the molecular mechanisms involved in RCC tumorigenesis and the development of new therapeutic targets are urgently needed to improve RCC therapy. Combination therapy with miRNAs and anticancer agents can be effective for achieving a potent drug response when the miRNAs function as tumor suppressors by targeting several essential oncogenes [9].

Dysregulation of miRNAs has been reported to have important roles during RCC tumorigenesis [10,11,12,13,14,15]. MiR-376a and miR-186 have been reported to be downregulated in RCC cells and their ectopic expression has been shown to dramatically suppress tumor proliferation [12,16]. In addition, we previously reported that miR-148a and miR-708 strongly induced apoptosis through caspase activation as well as by improving the sensitivities to a broad spectrum of anticancer drugs, including cisplatin and TNF-related apoptosis-inducing ligand (TRAIL), in RCCs [17,18].

MiR-1208 is encoded on chromosome 8q24, where chromosomal abnormalities are associated with the development of solid tumors [19,20,21]. In addition, miR-1208 is expressed at low levels in Burkitt’s lymphoma, breast cancer, and colon cancer cell lines [22]. Controversially, miR-1208 was shown to be upregulated in approximately 50% of gastric cancers when compared to the expression level in adjacent non-neoplastic tissues [23]. However, the clinical significance of miR-1208 and the molecular mechanism underlying its activity remain incompletely described in RCC. Therefore, in this study, we investigated the underlying molecular mechanism of miR-1208 and its capability for improving cisplatin sensitivity in RCC.

## 2. Results

### 2.1. MiR-1208 Inhibits Cell Proliferation and Promotes Apoptosis in Caki-1 Cells

To determine the growth-inhibitory effect of miR-1208, Caki-1 cells were treated with miR-Cont, miR-1208, and anti-miR-1208 followed by MTT assays to analyze cell viability. As shown in Figure 1A, the growth of Caki-1 cells was inhibited by miR-1208 in a time-dependent manner compared to the growth of miR-Cont-transfected Caki-1 cells. The population of sub-G1-phase cells markedly increased in the miR-1208-transfected Caki-1 cells compared to that in the miR-Cont-transfected cells (Figure 1B). On that basis, we next investigated whether transfection with miR-1208 resulted in caspase activation in Caki-1 cells. In addition, we assessed the effects on DNA fragmentation, which are hallmarks of apoptosis. MiR-1208 induced DNA fragmentation, while each compound treated separately was without effect (Figure 1C). Ectopic expression of miR-1208 led to a decrease in the protein levels of procaspase-3 and to an increase in the cleavage of poly ADP ribose polymerase (PARP) in Caki-1 cells at 24 h after transfection (Figure 1D).

### 2.2. MiR-1208 Reduces TBCK Expression by Directly Targeting Its 3′-UTR

A bioinformatic analysis program, TargetScan (http://www.targetscan.org), was used to identify the putative binding gene of miR-1208. The TargetScan miRNA target predictions showed that the TBCK 3′-untranslated region (3′-UTR) contained two potential binding sites for miR-1208 at the 215–222 and 2878–2884 nucleotide positions (http://www.targetscan.org/cgi-bin/targetscan/vert_72/view_gene.cgi?rs=ENST00000273980.5&taxid=9606&members=miR-1208&showcnc=1&shownc=1&shownc_nc=1&showncf1=1&showncf2=1&subset=1) (Figure 2A). To investigate whether exogenous miR-1208 could inhibit the expression of TBCK, Caki-1 cells were transiently transfected with miR-1208 or miR-Cont as a control for 24 h. TBCK expression was determined by performing RT-PCR and Western blot assays. As shown in Figure 2B,C, ectopic expression of miR-1208 reduced both TBCK mRNA and protein levels. In contrast, transfection with anti-miR-1208 resulted in an increase in TBCK expression in Caki-1 cells (Figure 2B,C). Subsequently, we examined whether miR-1208 could directly bind to the 3′-UTR of TBCK in renal cancer cells. Previously, we predicted that miR-1208 could bind to two different regions of the 3′-UTR of TBCK mRNA (Figure 2A). Thus, we investigated whether the miR-1208 was bound to either of those sites. The predicted miRNA binding sequences of TBCK (sites #1 and #2) were cloned into the downstream region of a luciferase reporter construct (pmirGLO-TBCK #1 and pmirGLO-TBCK #2, Figure 3A). Caki-1 cells were transiently transfected with either pre-miR-1208 or miR-Cont. As shown in Figure 3B, miR-1208 significantly reduced luciferase activity in both pmirGLO-TBCK #1 and #2 compared to that in miR-Cont. This result shows that miR-1208 specifically binds to both regions in the 3′-UTR of TBCK, thereby inhibiting the expression of TBCK. In contrast, the luciferase activity of the reporter vector containing the mutated 3′-UTR in TBCK was unaffected by miR-1208 (Figure 3C).

### 2.3. MiR-1208 Sensitizes Cisplatin-Induced Apoptosis in Renal Cancer Cells

Since TBCK has an important role in cell proliferation and cell growth through its capacity to enhance the mTOR pathway [24], we investigated whether the reduction of TBCK expression by miR-1208 could increase sensitivity to drugs that induce apoptosis in Caki-1 cells. MiR-1208-transfected cell lines were treated with cisplatin, and cytotoxicity was examined by performing FACS analysis. As shown in Figure 4A, transfection with miR-1208 resulted in a significant increase in the fraction of cells in the sub-G1 phase compared to the fraction in miR-Cont-transfected cells following treatment with cisplatin. Treatment of miR-1208-transfected cells with cisplatin also resulted in the cleavage of both PARP and procaspase-3 (Figure 4B). In contrast, the anti-miR-1208-transfected cells treated with cisplatin showed a decrease in the sub-G1 population and blocking of PARP and procaspase-3 cleavages compared to those in miR-Cont-transfected cells (Figure 4A,B). As shown in Figure 4C,D, cisplatin treatment in miR-1208-transfected cells increased the number of TUNEL-positive cells, but the cisplatin treatment in the anti-miR-1208-transfected cells reduced the number of TUNEL-positive cells compared to that in miR-Cont-transfected cells treated with cisplatin. Treatment of miR-1208-transfected cells with cisplatin induced DNA fragmentation compared to that in miR-Cont-transfected cells (Figure 4E).

### 2.4. MiR-1208 Sensitizes TRAIL-induced Apoptosis in Renal Cancer Cells

We next examined whether miR-1208 could increase the sensitivity of renal cancer cells to TRAIL. TRAIL treatment of miR-1208-transfected cells caused an increase in the cell fraction at the sub-G1 stage compared to the level in the miR-Cont-transfected cells (Figure 5A). As shown in Figure 5B, TRAIL treatment of miR-1208-transfected cells markedly led to procaspase-3 and PARP protein cleavages. Moreover, TRAIL treatment in miR-1208-transfected cells increased the number of TUNEL-positive cells (Figure 5C,D), whereas the TRAIL treatment of the anti-miR-1208-transfected cells reduced the number of TUNEL-positive cells compared to that in the miR-Cont plus TRAIL-treated cells. In addition, treatment of miR-1208-transfected cells with TRAIL induced greater DNA fragmentation than that in miR-Cont-transfected cells (Figure 5E).

### 2.5. MiR-1208 Plus Cisplatin- and MiR-1208 Plus TRAIL-Induced Apoptosis were Involved in Activation of Caspase-Dependent Apoptotic Pathways

Next, we examined whether activation of the caspase pathway has an important role in miR-1208 plus cisplatin-induced apoptosis. The miR-1208 plus cisplatin-induced apoptosis was prevented by pre-treatment with a caspase inhibitor, z-VAD-fmk, as shown by the results of the FACS analysis (Figure 6A,B). In addition, z-VAD-fmk treatment prevented cleavage of both procaspase-3 and PARP (Figure 6C). In addition, the miR-1208 plus TRAIL-induced apoptosis was prevented by pre-treatment with z-VAD-fmk, as determined by FACS and Western blot analyses (Figure 6D–F). These results suggest that apoptosis mediation induced by miR-1208 plus cisplatin or miR-1208 plus TRAIL is involved in activation of the caspase-dependent apoptotic pathway.

### 2.6. TBCK Knockdown Improves the Sensitivities to Cisplatin and TRAIL

To determine whether the anticancer effects of miR-1208 in Caki-1 cell were due to TBCK inhibition or to interaction with another gene, Caki-1 cells were transiently transfected with a small interfering RNA (si-RNA) specific to TBCK (si-TBCK) or with a scrambled si-RNA negative control (si-Cont). The si-TBCK knocked down the expression of TBCK protein in Caki-1 cells (Figure 7B,D). Depletion of TBCK by si-RNA markedly increased the sensitivity of the cells to apoptosis-inducing drugs, including cisplatin or TRAIL (Figure 7A,C).

## 3. Discussion

Our study has demonstrated that miR-1208 induces apoptosis in Caki-1 cells by activating the caspase pathway. In addition, transfection with miR-1208 was shown to enhance the sensitivity of Caki-1 cells to cisplatin and TRAIL. Moreover, we showed that ectopic expression of miR-1208 decreased TBCK mRNA and protein levels in Caki-1 cells, which were the result of direct binding of miR-1208 to the miR-1208 binding sites of the TBCK 3′-UTR. Finally, we have established the possibility that TBCK has anti-apoptotic activity as demonstrated by the si-RNA-mediated knockdown of TBCK in renal cancer cells.

The function of miR-1208 in cancer is controversial. MiR-1208 was upregulated in gastric cancers in relation to that in adjacent non-neoplastic tissues and in exocrine pancreatic secretions from pancreatic ductal adenocarcinoma (PDAC) patients [23,25], raising the possibility that miR-1208 might act as an oncogene. Controversially, miR-1208 was expressed at low levels in Burkitt’s lymphoma, breast cancer, and colon cancer cell lines [22]. Plasmacytoma variant translocation 1 (PVT1) is located on chromosome 8q24 and produces a wide variety of spliced non-coding RNAs and a cluster of six annotated miRNAs, miR-1204, miR-1205, miR-1206, miR-1207-5p, miR-1207-3p, and miR-1208 [26,27]. Treatment of HCT116 cells with daunorubicin resulted in the induction of endogenous PVT1 transcripts and all of the primary miRNAs at this locus [26]. These studies suggest that miR-1208 may act as a tumor suppressor in cancer cells. Consistent with results in a previous study, we found that miR-1208 induces apoptosis and activates the caspase-dependent pathway, indicating that it might function as a tumor suppressor in renal cancer cells. Next, we investigated the effects of miR-1208 on the sensitivities to anticancer drugs such as cisplatin and TRAIL. Introduction of miR-1208 increased the sensitivities to both cisplatin and TRAIL, suggesting that miR-1208 can promote sensitivities to anticancer drugs in Caki-1 cells.

Current challenges and strategies for miRNA-based cancer therapy are to achieve specific, efficient, and safe systemic delivery of therapeutic miRNAs in vivo. So far, the strategies employed in miRNA delivery for cancer therapy include the development of carriers for miRNA-based therapy that do not induce toxicity. The first strategy developed for delivery of miRNAs into tumors was the synthesis of chemically modified miRNAs [28]. Another approach was the design of nanoparticle preparations for manual diffusion in tumor tissues [29]. The third approach was to add surface modifications to the nanoparticles to allow specific binding to target cancer cells and to promote internalization of nanoparticles into cancer cells via receptor-mediated endocytosis [30]. Recently, systemic delivery of LNA-anti-miR-21 combined with cisplatin treatment over seven weeks completely abolished the onset of lung adenomas and hyperplasia [31]. In addition, in vivo delivery of miR-96 or miR-182 with cisplatin on nanoparticles that were embedded in a hydrogel scaffold patch had enabled significant primary tumor shrinkage and metastasis prevention [32]. In the present study, Caki-1 cells were treated with the anticancer drug cisplatin or TRAIL after miR-1208 transfection, but further studies are required to develop a smart delivery vehicle that can carry these types of anticancer agents into renal cancer cells in vitro and in vivo.

In the present study, we showed that TBCK is a direct target of miR-1208 and that part of the anticancer effects of miR-1208 against Caki-1 cells might be mediated through the downregulation of TBCK expression. TBCK has recently been implicated in the regulation of mammalian target of rapamycin (mTOR) signaling [24]. Therefore, it is necessary to identify the upstream regulators of TBCK that function in the suppression of tumor growth and enhancement of drug susceptibility. Previous studies have shown that si-RNA-mediated TBCK silencing decreased cell size and inhibited cell proliferation in HEK293 cells [24]. Our data showed that TBCK silencing by si-RNA enhances the sensitivity to cisplatin or TRAIL in Caki-1 cells. Moreover, miR-1208 directly targets TBCK by interacting with two binding sites in the 3′-UTR, which is involved in miR-1208-induced apoptosis and, in turn, increases susceptibility to cisplatin or TRAIL in kidney cancer cells. To the best of our knowledge, this is the first study to show that TBCK might work as an anti-apoptotic factor in Caki-1 cells.

In conclusion, the present study showed that TBCK was a direct target of miR-1208, and that miR-1208/TBCK interaction has an important role in the regulation of apoptosis as well as in the enhancement of cisplatin or TRAIL sensitivities in renal cancer cells. Thus, miR-1208 could be considered as a potential target for renal cancer therapy.

## 4. Materials and Methods

### 4.1. Cells and Materials 

Caki-1 cells were obtained from the American Type Culture Collection (Rockville, MD, USA). Caki-1 cells cell lines were maintained in Dulbecco’s modified Eagle’s medium (DMEM, Welgene, Daegu, Korea) supplemented with 20 mM HEPES buffer, and 100 μg/mL gentamicin. The antibody against PARP was from Cell Signaling Technology, Inc. (Boston, MA, USA). Antibodies against caspase-3, -TBCK, and -actin antibodies were purchased from Santa Cruz Biotechnology Inc. (Santa Cruz, CA, USA). Cisplatin was obtained from Sigma Chemical Co. (St. Louis, MO, USA). Soluble recombinant TRAIL was purchased from R&D Systems (Minneapolis, MN, USA). 

### 4.2. Western Blot Analysis

Cellular lysates were extracted using lysis buffer (137 mM NaCl, 15 mM EGTA, 0.1 mM sodium orthovanadate, 15 mM MgCl_2_, 0.1% Triton X-100, 25 mM MOPS, 100 μM phenylmethylsulfonyl fluoride, and 20 μM leupeptin, adjusted to pH 7.2). After being quantified using the BCA Protein Assay Kit (Pierce, Appleton, WI), the proteins samples was separated by SDS-PAGE, electrotransferred to Immobilon-P membranes (Millipore, Bedford, MA, USA), blocked, and incubated with primary antibodies. After being incubated with corresponding secondary antibodies including rabbit IgG-HRP or mouse IgG-HRP(Molecular Probes, Eugene, OR, USA), the blots developed with ECL Western blotting kit (Millipore) according to the manufacturer’s instructions.

### 4.3. Cell Viability Assay

Cell viability was assessed using the WelCount Cell Viability Assay kit (Welgene). Briefly, Caki-1 cells were seeded in 96-well plate with 2.5 × 10^4^ cells per well. After transfecting with miRNAs for 24, 48, or 72 h, MTT reagents was added to medium and incubated 2 h at room temperature. Absorbance was tested at 450 nm using a microplate spectrophotometer (Thermo Scientific, Waltham, MA, USA) at 450/690 nm.

### 4.4. Flow Cytometry Analysis

After Caki-1 were trypsinized and harvested by centrifugation, approximately 1 × 10^6^ Caki-1 cells were suspended in 100 μL of PBS. The cells were fixed in 200 μL of 95% ethanol at 4 °C for 1 h, washed with PBS, and resuspended in 250 μL of RNase solution. After incubation 37 °C for 30 min, the cellular DNA was then stained with 250 μL of propidium iodide (50 μg/mL) for 30 min at room temperature. The stained cells were subjected to FACScan flow cytometer to determine the relative DNA content based on red fluorescence (BD Biosciences, San Jose, CA, USA).

### 4.5. RNA Isolation and Reverse Transcriptase-Polymerase Chain Reaction (RT-PCR)

Total RNA was isolated by easy-BLUE™ Total RNA Extraction Kit (Intron, Sungnam, Korea) according to the manufacturer’s instructions. Next, 5 μg of total RNA reverse-transcribed by M-MLV reverse transcriptase (Promega, Madison, WI, USA). The cDNAs for TBCK and actin were amplified by performing PCR with specific primers. The sequences of the upstream and downstream primers used are as follows: *TBCK* 5′-TGCACACTCCCCAATTTTCT-3′ and 5′-TAGCTCCCTCAACTCCCAGA-3′, *actin* 5′-AGCAAGAGAGGCATCCTCACCCTGAAGTAC-3′ and 5′-GCACAGCTTCTCCTTAATGTCACGCACGAT-3′. The amplified PCR products were electrophoretically resolved on agarose gel and were visualized by performing ethidium bromide staining.

### 4.6. Luciferase Reporter Assays

The 3′-UTR region of TBCK or mutated 3′-UTR region of TBCK were cloned into a pmirGLO Dual-Luciferase reporter plasmid (Promega). Caki-1 cells were transfected with the constructed TBCK 3′-UTR-pmirGLO Dual-Luciferase reporter plasmid, miR-1208, or anti-miR-1208 (Genolution, Seoul, Korea) by Lipofectamin RNAiMAX Reagent (Invitrogen, Carlsbad, CA, USA). Twenty-four hours post-transfection, luciferase activity assays were performed and results normalized to Renilla luciferase activity (Promega). The experiments were repeated three times.

### 4.7. Terminal Deoxynucleotidyl Transferase-Mediated dUTP Fluorescein Nick end Labeling (TUNEL) Assay

Terminal deoxynucleotidyl transferase (TdT) dUTP nick-end labeling (TUNEL) staining was performed with an in-situ cell death detection kit (Roche, Indianapolis, IN, USA) in accordance with the manufacturer’s protocol. The condensed or fragmented nuclei of apoptotic cells were visualized with a Leica DMI6000B fluorescence microscope (Leica Biosystems, Wetzlar, Germany). All measurements were performed in a blinded manner, and at least three independent experiments were conducted

### 4.8. Cell Death Assessment by DNA Fragmentation Assay 

A cell death detection ELISA plus kit (Roche) was used for assessing apoptotic activity based on the detection of fragmented DNA. After cells were grown in 96-well plates and transfected with miR-Cont, miR-1208, or anti-miR-1208 inhibitor, cells were treated with cisplatin or TRAIL. Culture plate was centrifuged for 10 min at 200× *g*, the supernatant was removed, and the pellet was lysed for 30 min. The plate was centrifuged again at 200× *g* for 10 min, and the supernatant fraction containing the cytoplasmic histone-associated DNA fragments was harvested and incubated with anti-histone antibody. The absorbance of each well was measured using microplate reader at 405 and 490 nm (reference wavelength). The signals in the wells containing the substrate alone were subtracted as background signals.

### 4.9. Statistical Analysis

All experiments were repeated at least three times independently, and comparisons between the experimental and control groups were conducted. Statistical significance was measured using Student’s *t*-test. The * indicates a *P*-value < 0.05, which was considered statistically significant. 

## Figures and Tables

**Figure 1 ijms-20-03540-f001:**
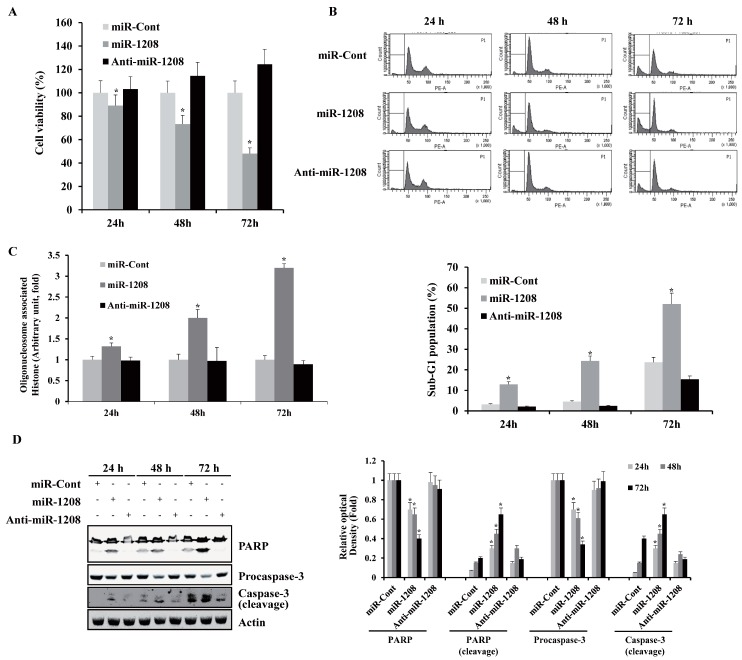
MiR-1208 inhibits renal cancer cell proliferation and promotes apoptosis. (**A**) Caki-1 cells were transfected with the indicated miRNAs for 24, 48, or 72 h and their cell viability was quantified using the MTT assay. The data are reported as the mean ± SD (*n* = 3). * indicates significantly different from miR-Cont-transfected cells (*P* < 0.05). (**B**) Caki-1 cells were transfected with miR-1208 or miR-Cont for 24, 48, or 72 h. FACS analysis was performed to measure a sub-G1 fraction. Histograms of flow cytometry are shown in the **upper panel**. Percentages of sub G1 populations are shown in **bottom panel**. Data are reported as the mean ± SD (*n* = 3). * indicates significantly different from miR-Cont-transfected cells (*P* < 0.05). (**C**) Nuclei and fragmented DNA were detected by a DNA fragmentation detection kit* indicates significantly different from miR-Cont-transfected cells (*P* < 0.05). (**D**) Equal amounts of cell lysates (50 μg) were subjected to electrophoresis and were immunoblotted for PARP and caspase-3 antibodies. Actin was used for normalization in all immunoblots. Using ImageJ software, the intensity of each protein in miRs-transfected cells were normalized to actin and expressed as a ratio of the densitometric value (**right panel**). * indicates significantly different from miR-Cont-transfected cells (*P* < 0.05) at each time point.

**Figure 2 ijms-20-03540-f002:**
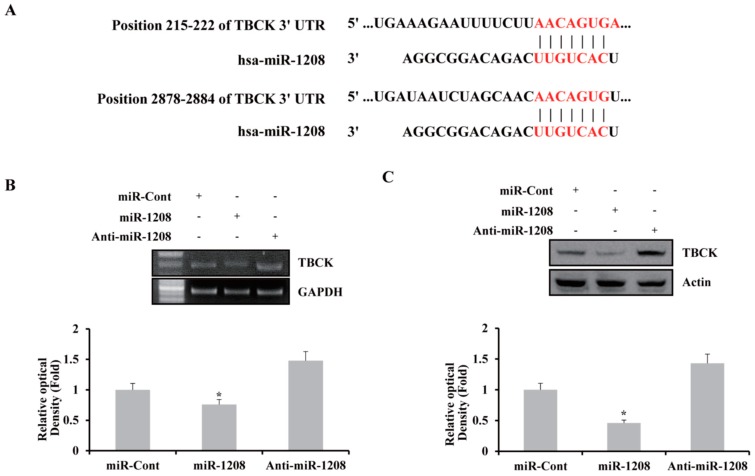
Identification of the gene regulated by miR-1208 in renal cancer cells. (**A**) miR-1208-binding sites in the 3′-UTR of TBCK mRNA. (**B**) RT-PCR analysis of TBCK mRNA expression in the miR-Cont- or miR-1208- or anti-miR-1208-transfected Caki-1 cells. The relative levels of each TBCK mRNA are expressed as the ratio of the densitometric value of each band to that of GAPDH (**bottom**
**panel**). * indicates *P* < 0.05 compared to miR-Cont-transfected cells. (**C**) Immunoblots for TBCK protein in the miR-Cont or miR-1208 or anti-miR-1208 transfected Caki-1 cells. Actin was used as the loading control. The relative levels of each TBCK protein in miRNA-transfected cells are expressed as the ratio of the densitometric value of each band to that of actin (**bottom**
**panel**). * indicates *P* < 0.05 compared to miR-Cont-transfected cells.

**Figure 3 ijms-20-03540-f003:**
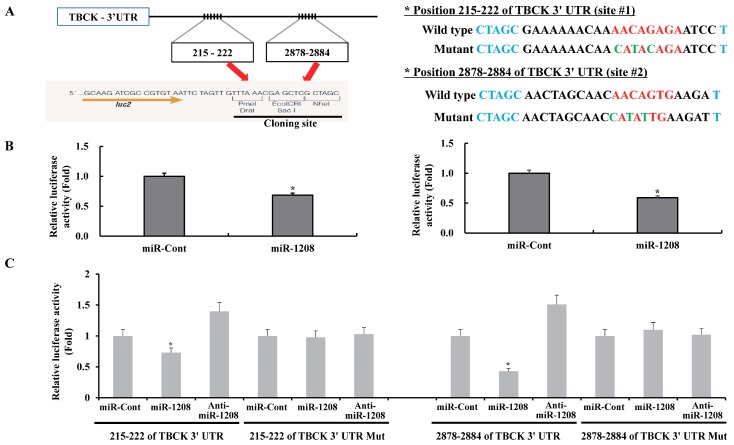
TBCK is a functional target of miR-1208 in renal cancer. (**A**) Schematic representation of TBCK showing putative miR-1208 target sites (**left panel**). The mutated nucleotides in the TBCK 3′-UTR mutant (**right panel**). (**B**) Luciferase activity assay with the corresponding wild-type luciferase constructs containing two different miR-1208 target sequences (site #1 or site #2) transfected with miR-Cont or miR-1208. For all luciferase activity assays, firefly luciferase values were normalized to Renilla luciferase activity and plotted as the relative luciferase activity. Data are reported as the mean ± SD (*n* = 3). Student’s *t*-test for unpaired values was used. * indicates *P* < 0.05 compared to miR-Cont-transfected cells. (**C**) Luciferase assays showing the repression of wild-type UTR (TBCK 3′-UTR) or mutant UTR (TBCK 3′UTR-mutant), following the transfection of miR-1208, anti-miR-1208, and miR-Cont. Data are shown as the mean ± SD (*n* = 3). * indicates *P* < 0.05 compared to miR-Cont-transfected cells.

**Figure 4 ijms-20-03540-f004:**
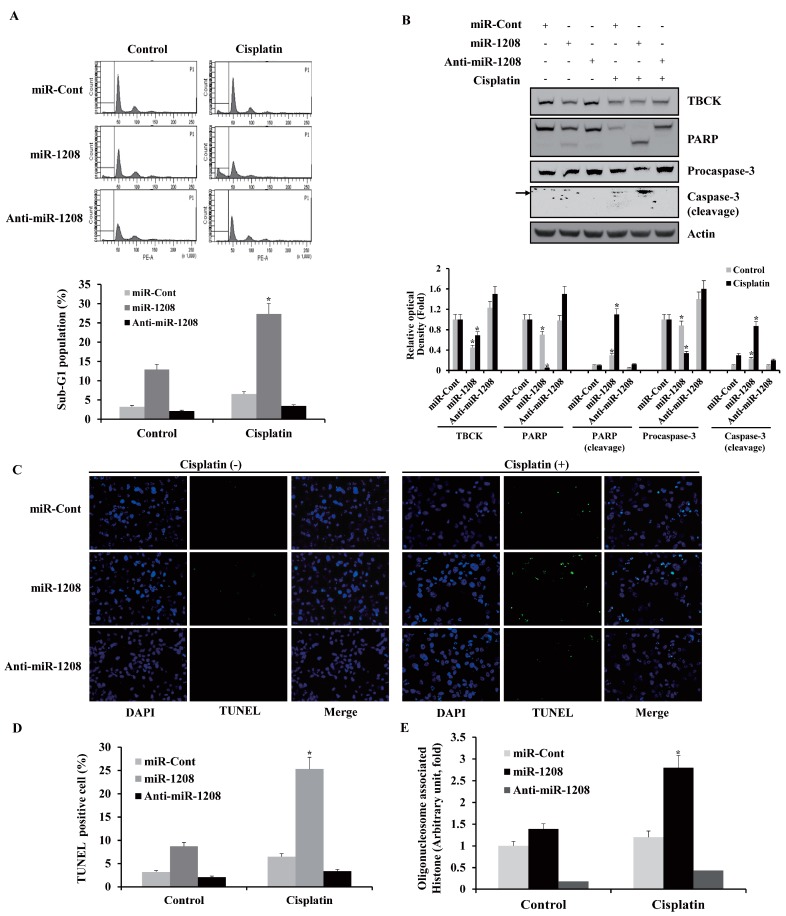
MiR-1208 sensitizes renal cancer cells to cisplatin-mediated apoptosis. (**A**) After Caki-1 cells were transfected with miR-1208 or miR-Cont for 24 h, cells were treated with cisplatin for 24 h. FACS analysis was performed to measure a sub-G1 fraction. Histograms of flow cytometry are shown in the **upper panel**. Percentages of sub G1 populations are shown in **bottom panel**. Data are reported as the mean ± SD (*n* = 3). * indicates *P* < 0.05 compared to miR-Cont-transfected cells. (**B**) Equal amounts of cell lysates (50 μg) were loaded to electrophoresis and were immunoblotted for PARP and caspase-3 antibodies. Actin was used for normalization in all immunoblots. Using ImageJ software, the intensity of each protein in miRNA and cisplatin-treated cells were normalized to actin and expressed as a ratio of the densitometric value (**bottom panel**). * indicates *P* < 0.05 compared to miR-Cont-transfected cells. (**C**) After Caki-1/miR-1208 and Caki-1/miR-Cont cells were treated with cisplatin, TUNEL analysis was performed. TUNEL-stained cells were observed by fluorescence microscopy at ×200 magnification. TUNEL-positive cells are shown in green. * indicates *P* < 0.05 compared to miR-Cont-transfected cells. See high-resolution images in Appendix A. (**D**) The graphs show numbers of TUNEL-positive cells. * indicates *P* < 0.05 compared to miR-Cont-transfected cells. (**E**) Nuclei and fragmented DNA were detected by using a DNA fragmentation detection kit. * indicates *P* < 0.05 compared to miR-Cont-transfected cells.

**Figure 5 ijms-20-03540-f005:**
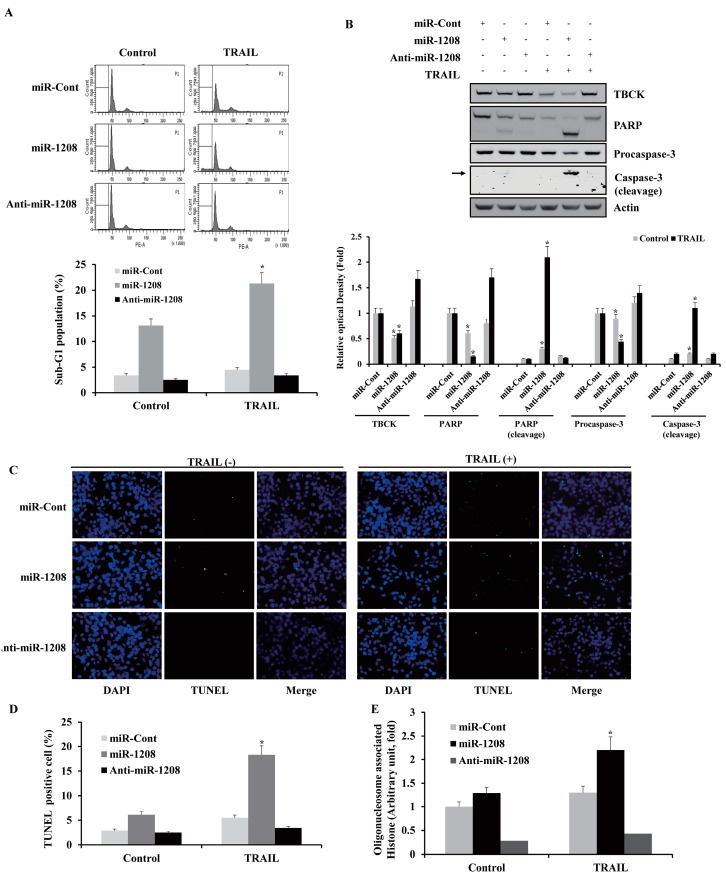
MiR-1208 sensitizes renal cancer cells to TRAIL-mediated apoptosis. (**A**) After Caki-1 cells were transfected with miR-1208 or miR-Cont for 24 h, cells were treated with TRAIL for 24 h. FACS analysis was performed to measure a sub-G1 fraction. Histograms of flow cytometry are shown in the **upper panel**. Percentages of sub G1 populations are shown in **bottom panel**. Data are reported as the mean ± SD (*n* = 3). * indicates *P* < 0.05 compared to miR-Cont-transfected cells. (**B**) Equal amounts of cell lysates (50 μg) were loaded to electrophoresis and were immunoblotted for PARP and caspase-3 antibodies. Actin was used for normalization in all immunoblots. Using ImageJ software, the intensity of each protein in miRNA and TRAIL-treated cells were normalized to actin and expressed as a ratio of the densitometric value (**bottom panel**). * indicates *P* < 0.05 compared to miR-Cont-transfected cells. (**C**) After Caki-1/miR-1208 and Caki-1/miR-Cont cells were treated with TRAIL, TUNEL analysis was performed. TUNEL-stained cells were observed by fluorescence microscopy at ×200 magnification. TUNEL-positive cells are shown in green. * indicates *P* < 0.05 compared to miR-Cont-transfected cells. See high-resolution images in Appendix A. (**D**) The graphs show numbers of TUNEL-positive cells. * indicates *P* < 0.05 compared to miR-Cont-transfected cells. (**E**) Nuclei and fragmented DNA were detected by using a DNA fragmentation detection kit. * indicates *P* < 0.05 compared to miR-Cont-transfected cells.

**Figure 6 ijms-20-03540-f006:**
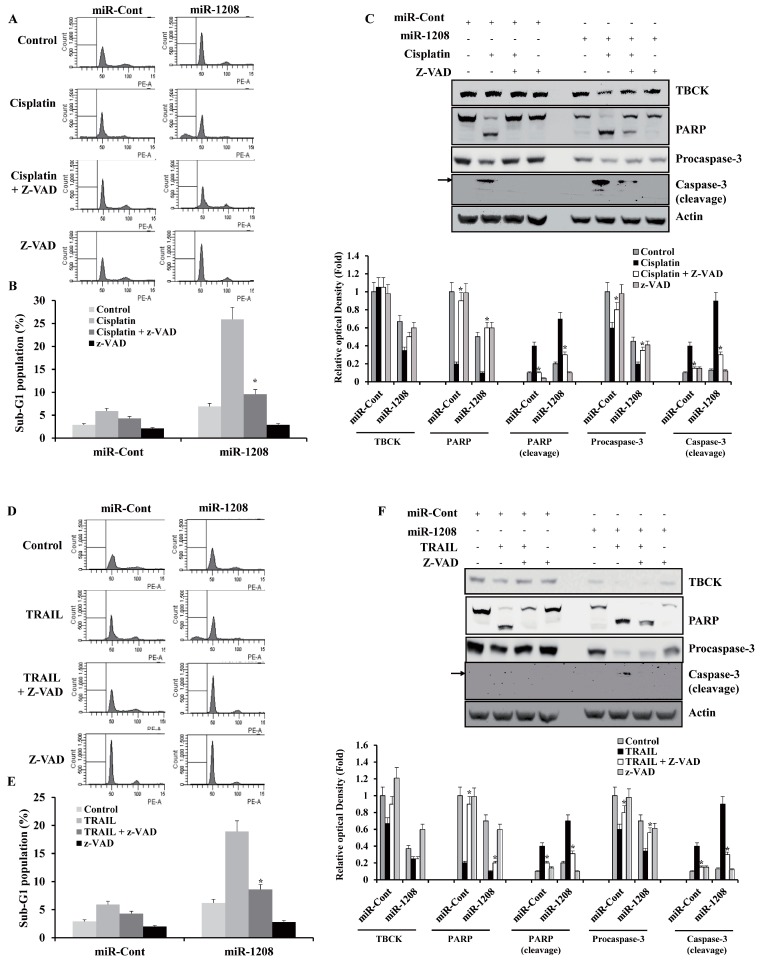
Ectopic expression of miR-1208 promotes cisplatin- or TRAIL-induced apoptosis through activation of caspase cascades. (**A**) Caki-1/miR-1208 cells were incubated with 50 μM z-VAD-fmk or solvent for 1 h before treatment with cisplatin for 24 h. FACS analysis was performed to measure a sub-G1 fraction. Histograms of flow cytometry are shown. **(B)** Percentages of sub G1 populations are shown. Data are reported as the mean ± SD (*n* = 3). * indicates *P* < 0.05 versus cisplatin-treated cells. (**C**) Equal amounts of cell lysates (50 μg) were loaded to electrophoresis and were immunoblotted for PARP and caspase-3 antibodies. Actin was used for normalization in all immunoblots. Using ImageJ software, the intensity of each protein in miRNA and cisplatin-treated cells were normalized to actin and expressed as a ratio of the densitometric value (**bottom panel**). * indicates *P* < 0.05 versus cisplatin-treated si-Cont- or miR-1208-transfected cells. (**D**) Caki-1/miR-1208 cells were incubated with 50 μM z-VAD-fmk or solvent for 1 h before treatment with TRAIL for 24 h. FACS analysis was performed to measure a sub-G1 fraction. Histograms of flow cytometry are shown. (**E**) Percentages of sub G1 populations are shown. Data are reported as the mean ± SD (*n* = 3). * indicates *P* < 0.05 versus TRAIL-treated cells. (**F**) Equal amounts of cell lysates (50 μg) were loaded to electrophoresis and were immunoblotted for PARP and caspase-3 antibodies. Actin was used for normalization in all immunoblots. Using ImageJ software, the intensity of each protein in miRNA and TRAIL-treated cells were normalized to actin and expressed as a ratio of the densitometric value (**bottom panel**). * indicates *P* < 0.05 versus TRAIL-treated si-Cont- or miR-1208-transfected cells.

**Figure 7 ijms-20-03540-f007:**
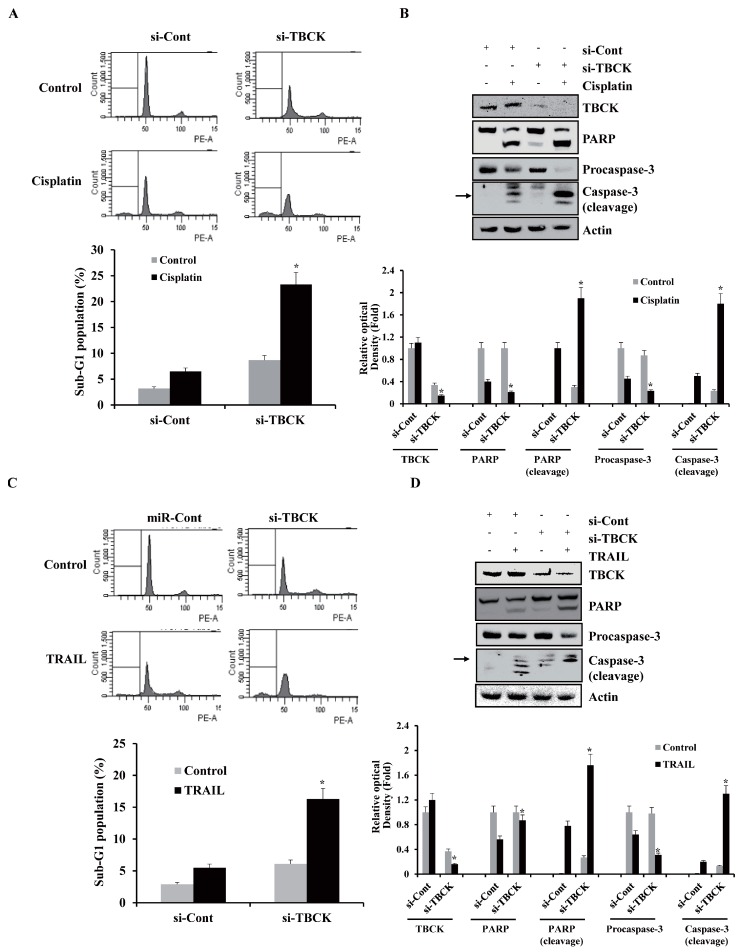
si-TBCK-mediated TBCK downregulation increases the sensitivities to cisplatin or TRAIL. (**A**) Caki-1 cells were transfected with si-Cont or si-TBCK. After 24 h transfection, the cells were treated with cisplatin for 24 h. FACS analysis was performed to measure a sub-G1 fraction. Histograms of flow cytometry are shown in the **upper panel**. Percentages of sub G1 populations are shown in **bottom panel**. * indicates *P* < 0.05 compared to cisplatin-treated si-Cont-transfected cells. (**B**) Equal amounts of cell lysates (50 μg) were loaded to electrophoresis and were immunoblotted for PARP and caspase-3 antibodies. Actin was used for normalization in all immunoblots. Using ImageJ software, the intensity of each protein in miRNA and cisplatin-treated cells were normalized to actin and expressed as a ratio of the densitometric value (**bottom panel**). * indicates *P* < 0.05 compared to non-treated si-Cont-transfected cells. (**C**) Caki-1 cells were transfected with si-Cont or si-TBCK. After 24 h transfection, the cells were treated with TRAIL for 24 h. FACS analysis was performed to measure a sub-G1 fraction. Histograms of flow cytometry are shown in the **upper panel**. Percentages of sub G1 populations are shown in **bottom panel**. * indicates *P* < 0.05 compared to TRAIL-treated si-Cont-transfected cells. (**D**) Equal amounts of cell lysates (50 μg) were loaded to electrophoresis and were immunoblotted for PARP and caspase-3 antibodies. Actin was used for normalization in all immunoblots. Using ImageJ software, the intensity of each protein in miRNA and TRAIL-treated cells were normalized to actin and expressed as a ratio of the densitometric value (**bottom panel**). * indicates *P* < 0.05 compared to non-treated si-transfected Cont cells.

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
