# Peer review of "MiR-1208 Increases the Sensitivity to Cisplatin by Targeting TBCK in Renal Cancer Cells"

_ijms, 2019, doi:10.3390/ijms20143540_

Reviewer 1 Report

The authors investigated the mechanisms of miRNA-1208 on the renal cell carcinoma in vitro combined with cisplatin treatment. The figures and presentation of the work need to be improved. 

The number labeling of each section is not correct. The Results should be part 2. All the sub-sections of Results are labeled as 1.1. 

Please provide higher resolution of figures. It seems to me that only figure 1 has the publishable resolution. Please also check all the words on the figures. Some look weird, for example, figure 6E. 

Figure 1 contains a “Fig 1” on the bottom right of the figure. 

Figure 2 and 3 are very confusing because there is only one figure but two captions for that figure. 

Figure 4 and 5 do not have label C in the figure panel. 

Figure 7 does not have label A in the figure panel.

I would recommend “revise the manuscript for publication”. 

Author Response

Respond to Reviewer’s comments

Language of the manuscript should be thoroughly edited to increase the readability.

Response: As the reviewer pointed out, this manuscript was edited by “NURISCO (http://www.nurisco.net/company/introduction.do)”. Please refer to the Certificate at the end of the response letter.

Reviewer 1

We thank the reviewer for his/her comments on our work.

1. The number labeling of each section is not correct. The Results should be part 2. All the sub-sections of Results are labeled as 1.1.

     Response: As reviewer point out, we corrected number labeling of each section.

 2. Please provide higher resolution of figures. It seems to me that only figure 1 has the publishable resolution. Please also check all the words on the figures. Some look weird, for example, figure 6E.

    Response: As reviewer point out, we provided high-resolution of Figures.

 3. Figure 1 contains a “Fig 1” on the bottom right of the figure.

    Response: As reviewer point out, “Fig.1” was deleted.

 4. Figure 2 and 3 are very confusing because there is only one figure but two captions for that figure.

    Response: As reviewer point out, we provided Fig.2 and Fig. 3.

 5. Figure 4 and 5 do not have label C in the figure panel.

    Response: As reviewer point out, we corrected the label in Fig.4 and Fig.5.

 6. Figure 7 does not have label A in the figure panel.

    Response: As reviewer point out, we corrected the label in Fig.7.

Reviewer 2 Report

The authors of this article entitled, ‘MiR-1208 increases the sensitivity to cisplatin by targeting TBCK in renal cancer cells’ clearly shows the involvement of MiR-1208 in renal cancer cell death. However, there are some concerns about the data representation, experiments and the conclusions derived from it. Some of the concerns are listed below:

1.     What is the basal level expression of MiR-1208 in Caki-1 cells as compare to normal cell line?

2.     There is a discrepancy in the sub-G1 population (%) and western blot data provided at 24hr time point. There is no clear evidence of total number or percentage of apoptotic cells in different treatment groups. Is there any other form of cell death involve in sub-G1 population (%) count such as necrosis, necro-apoptosis, oncosis, etc? Annexin V apoptotic assay might help to address this question.

3.     All western blot data should be supplemented with quantitative analysis. There is a discrepancy in the statement and data provided. Example, Fig 1C. 24hr panel. There is no change at 24hr time point. Besides, there is no cleaved caspase 3. Consistent data representation is needed.

4.     Figure 2 and 3. Either one of the figures is missing (figure 2 maybe)

5.     All immunocytochemistry (TUNEL) images are not clear. Higher magnification images can be used for better representation.

6.     Statistics are not mentioned properly in the figure legends. It is not clear when and where ANOVA is used.

Author Response

Respond to Reviewer’s comments

 Language of the manuscript should be thoroughly edited to increase the readability.

Response: As the reviewer pointed out, this manuscript was edited by “NURISCO (http://www.nurisco.net/company/introduction.do)”. Please refer to the Certificate at the end of the response letter.

Reviewer 2

We thank the reviewer for his/her comments on our work.

The authors of this article entitled, ‘MiR-1208 increases the sensitivity to cisplatin by targeting TBCK in renal cancer cells’ clearly shows the involvement of MiR-1208 in renal cancer cell death. However, there are some concerns about the data representation, experiments and the conclusions derived from it. Some of the concerns are listed below:

What is the basal level expression of MiR-1208 in Caki-1 cells as compare to normal cell line?

Response: Thank you for your kind suggestion. We are collecting kidney cancer cell lines and normal kidney cell line as well as kidney cancer tissues and their normal tissues for the next project. In the next project, we will study the mutual regulatory mechanism between miRNA-1208 and TBCK.

2. There is a discrepancy in the sub-G1 population (%) and western blot data provided at 24hr time point. There is no clear evidence of total number or percentage of apoptotic cells in different treatment groups. Is there any other form of cell death involve in sub-G1 population (%) count such as necrosis, necro-apoptosis, oncosis, etc? Annexin V apoptotic assay might help to address this question.

Response: Thank you for your kind comments. We repeated Western blot and changed the Fig.1. We also added DNA fragmentation data, which is the hallmark of apoptosis.

3. All western blot data should be supplemented with quantitative analysis. There is a discrepancy in the statement and data provided. Example, Fig 1C. 24hr panel. There is no change at 24hr time point. Besides, there is no cleaved caspase 3. Consistent data representation is needed.

Response: As reviewer point out, the quantitative analysis for western blot data were added to All western blot data. Also, we provided Western blot for cleaved caspase-3 in Fig.1.

4. Figure 2 and 3. Either one of the figures is missing (figure 2 maybe).

Response: As reviewer point out, we provided Fig.2 and Fig. 3.

5. All immunocytochemistry (TUNEL) images are not clear. Higher magnification images can be used for better representation.

Response: We absolutely agree with your point. We are very sorry. The experiment using the fluorescence microscope is unavailable because of the mechanical problems of the instrument and time limit. Instead, we performed experiments using a DNA fragmentation assay kit. The results are shown in Fig.1, Fig. 4, and Fig.5. Also, we provided the enlarged size of the images to increase visibility in Supplementary Figures.

6. Statistics are not mentioned properly in the figure legends. It is not clear when and where ANOVA is used.

Response: As reviewer point out, we corrected error in the statistical description in Materials and Methods section.

Reviewer 3 Report

This manuscript describes the ability of small RNA molecules (miR-1208) to act as sensitizers of the anticancer drug cisplatin and the cytokine TRAIL in renal cancer cells. The investigators use a luciferase reporter assay to show that miR-1208 binds to two sites on the mRNA gene TBCK, which suppresses the expression of the gene. Their findings suggest that the tumor repressor properties of miR-1208 could be used to enhance the effects of cisplatin in cancer chemotherapy. This manuscript meets the standard set by the journal. Following minor revision, publication is recommended.

Comments/suggestions

The Discussion Section is very short. Some possible talking points for discussion would be to explain to the reader why cells that have been transfected with miR-1208 are treated with cisplatin after a delay of 24 h. What was the rational for choosing this time delay following the transfection with miR-1208? If chemotherapy involving both miR-1208 and cisplatin is ultimately envisioned, would a patient need to visit the clinic on two successive days for this combination chemotherapy? Certainly, cisplatin would directly react with miR-1208 if co-administration or only a brief delay between administration of the two agents takes place. A comment or two on the method of delivery of the microRNAs for possible chemotherapy would be helpful.

Author Response

Respond to Reviewer’s comments

 Language of the manuscript should be thoroughly edited to increase the readability.

Response: As the reviewer pointed out, this manuscript was edited by “NURISCO (http://www.nurisco.net/company/introduction.do)”. Please refer to the Certificate at the end of the response letter.

Reviewer 3

We thank the reviewer for his/her comments on our work.

This manuscript describes the ability of small RNA molecules (miR-1208) to act as sensitizers of the anticancer drug cisplatin and the cytokine TRAIL in renal cancer cells. The investigators use a luciferase reporter assay to show that miR-1208 binds to two sites on the mRNA gene TBCK, which suppresses the expression of the gene. Their findings suggest that the tumor repressor properties of miR-1208 could be used to enhance the effects of cisplatin in cancer chemotherapy. This manuscript meets the standard set by the journal. Following minor revision, publication is recommended.

Comments/suggestions

The Discussion Section is very short. Some possible talking points for discussion would be to explain to the reader why cells that have been transfected with miR-1208 are treated with cisplatin after a delay of 24 h. What was the rational for choosing this time delay following the transfection with miR-1208? If chemotherapy involving both miR-1208 and cisplatin is ultimately envisioned, would a patient need to visit the clinic on two successive days for this combination chemotherapy? Certainly, cisplatin would directly react with miR-1208 if co-administration or only a brief delay between administration of the two agents takes place. A comment or two on the method of delivery of the microRNAs for possible chemotherapy would be helpful.

Response: Thank you for your kind suggestion. We have supplemented Discussion sections for better understanding. We have added a short discussion on delivery for both miR-1208 and cisplatin.

 Response: We conducted a time delay experiment for the following reasons; It was reported that commercial transfection reagents are almost universally toxic because of their cationic or amphiphilic character[1]. Therefore, we conducted the experiment at time intervals for minimizing adverse effect of transfection reagents and extending the time for suppressing the expression of TBCK by miR-1208.

[1] Wang T, et al Molecules. 2018 Oct 8;23(10). pii: E2564; Lv H, et al. J Control Release. 2006 Aug 10;114(1):100-9; Breunig M, et al. Proc Natl Acad Sci U S A. 2007 Sep 4;104(36):14454-9. Epub 2007 Aug 28.

Round  2

Reviewer 1 Report

Please check the cell images in the figures. Some of them are clearly not focused well. Please re-take the images. After that, I agree for publication. 

Author Response

Respond to Reviewer’s comments

We thank the reviewer for his/her comments on our work.

Please check the cell images in the figures. Some of them are clearly not focused well. Please re-take the images. After that, I agree for publication.

Answer: We are very sorry for the inconvenience and confusion. The image was blurred while converting the original file. We changed the converting tool to provide clearer images (Fig. 4 & Fig.5). Also, we provided the enlarged size of the images to increase visibility in Supplementary Figs.

Reviewer 2 Report

Authors have addressed all comments. I think this research article in its current format is acceptable. 

Author Response

Respond to Reviewer’s comments

Authors have addressed all comments. I think this research article in its current format is acceptable. 

Response: We thank the reviewer for his/her comments on our work.

.
